# NOTCH Signaling in Osteosarcoma

**Zhenhao Zhang, Wei Wu * and Zengwu Shao ***

Department of Orthopedics, Union Hospital, Tongji Medical College, Huazhong University of Science and Technology, Wuhan 430022, China
* Correspondence: waynewu@hust.edu.cn (W.W.); 1985xh0536@hust.edu.cn (Z.S.)

**Abstract:** The combination of neoadjuvant chemotherapy and surgery has been promoted for the treatment of osteosarcoma; however, the local recurrence and lung metastasis rates remain high. Therefore, it is crucial to explore new therapeutic targets and strategies that are more effective. The NOTCH pathway is not only involved in normal embryonic development but also plays an important role in the development of cancers. The expression level and signaling functional status of the NOTCH pathway vary in different histological types of cancer as well as in the same type of cancer from different patients, reflecting the distinct roles of the Notch pathway in tumorigenesis. Studies have reported abnormal activation of the NOTCH signaling pathway in most clinical specimens of osteosarcoma, which is closely related to a poor prognosis. Similarly, studies have reported that NOTCH signaling affected the biological behavior of osteosarcoma through various molecular mechanisms. NOTCH-targeted therapy has shown potential for the treatment of osteosarcoma in clinical research. After the introduction of the composition and biological functions of the NOTCH signaling pathway, the review paper discussed the clinical significance of dysfunction in osteosarcoma. Then the paper reviewed the recent relevant research progress made both in the cell lines and in the animal models of osteosarcoma. Finally, the paper explored the potential of the clinical application of NOTCH-targeted therapy for the treatment of osteosarcoma.

**Keywords:** osteosarcoma; NOTCH; signaling; prognosis; molecular targeted therapy

## 1. Introduction

Osteosarcoma is the most common primary malignant tumor of bones [1]. Most osteosarcoma patients currently receive combinatorial treatment of doxorubicin, cisplatin, and methotrexate as the first-line therapy; however, local recurrence and lung metastasis rates remain high [2–4]. Despite the numerous trials conducted to evaluate novel therapies for metastatic osteosarcoma, the long-term survival of patients remains dismally bad [5–7]. Tyrosine kinase inhibitors such as regorafenib have been the major drug for treating metastatic osteosarcoma [8]. Other drug classes have been trialed for metastatic osteosarcoma based on promising pre-clinical data but have yielded generally disappointing outcomes [9,10]. Therefore, evaluating the potential of therapeutics targeting NOTCH for the treatment of osteosarcoma is of practical clinical importance. A potential relationship exists between the occurrence and progression of osteosarcoma and bone differentiation defects [11,12]. NOTCH signaling is an important mechanism regulating the normal development and differentiation of bone [13]. The complex and variable NOTCH pathway exerts different downstream effects and modulates cell fates [14]. The expression level and signaling functional status of the NOTCH pathway vary in different tumors and even in the same tumor, thus playing distinct roles [14]. However, the positive or negative effect of the pathway on cancers has not been identified clearly. Recent clinical trials have reported suitable efficacy for the treatment of osteosarcoma with the strategy of inhibition of the expression and function of the NOTCH pathway. To date, systematic reviews [15–17] published have not been concerned with the topic of the clinical significance of the alteration of the expression

and the dysfunction of the NOTCH pathway in osteosarcoma. Furthermore, the latest research advances in the NOTCH pathway in osteosarcoma were summarized in this paper.

## 2. Composition of the NOTCH Signaling Pathway

The NOTCH signaling pathway is composed of NOTCH ligand, NOTCH receptor, related enzymes, transcription factor CSL, regulatory factor, and NOTCH signaling downstream molecules [18]. NOTCH ligands, namely Delta/Serrate/Lag2 (DSL) family, belong to one-way transmembrane proteins. Mammals have five DSL ligands: Dll1, Dll4, and Dll3 are members of the delta-like ligand family; Jag1 and Jag2 are members of the serrate ligand family [18]. The NOTCH receptors, a series of transmembrane glycoproteins, are composed of extracellular regions, transmembrane regions, and cytoplasmic regions [19]. The NOTCH receptors (NOTCH1-4), encoded by different genes, differ in structures and can be degraded by a variety of proteases [19]. The cleaved NOTCH intracellular domain (NICD) is released into the cytoplasm and then transported to the nucleus to form the NOTCH transcription complex (NTC), which is composed of NICD, DNA binding factor, and transcriptional coactivators [19]. After combining with NOTCH regulatory element (NRE), NTC recruits transcription co-regulatory factors and starts the transcription of the specific target genes (such as Hes1, Hes5, etc.) [19]. The signaling cascade of the NOTCH pathway often begins with the interaction between the NOTCH receptor and the DSL ligand [19]. DSL ligands activate specific NOTCH receptors and induce their cleavage. Then NOTCH intracellular domain (NICD) is released and transported to the nucleus, thus forming NOTCH transcription complex (NTC) with transcription factor CSL, which acts as a transcription coactivator to start the transcription of NOTCH target genes [19] (Figure 1).

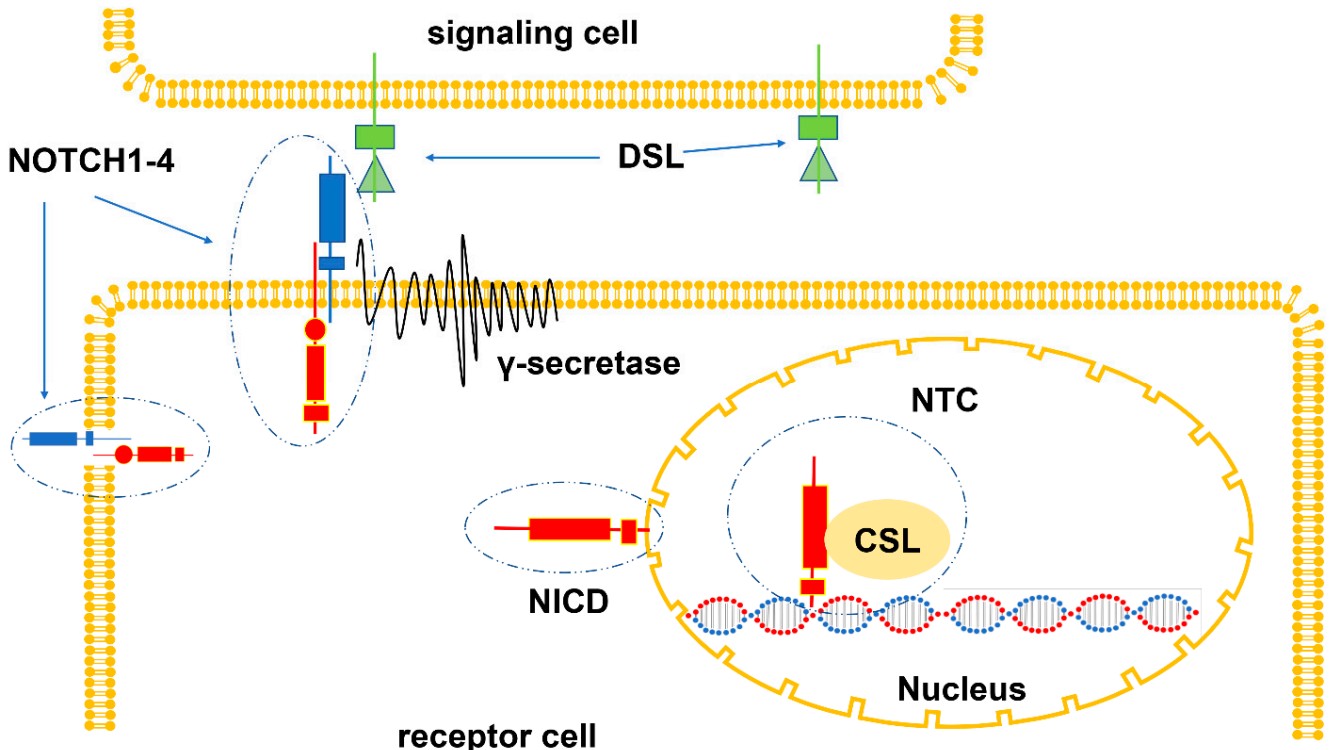

**Figure 1.** The Notch signaling pathway.

## 3. Biological Functions of the NOTCH Signaling Pathway

In mammals, NOTCH1 and NOTCH2 are widely expressed in most tissues, the highest expression of NOTCH3 is in vascular smooth muscle cells and pericytes, and that of NOTCH4 is in endothelial cells [14]. The NOTCH pathway modulates cell fate through adjusting signal intensities and dynamics as well as the diversity of ligand-receptor

binding [20–22]. Under physiological conditions, NOTCH signaling regulates and determines the fate of different tissues and cells during embryonic development [23]. In addition, the NOTCH signaling pathway has a profound effect on tumors. Activation of the NOTCH pathway can promote the occurrence and development of several malignancies, such as T-cell leukemia [24], pancreatic cancer [25] and colon cancer [26], among others. In contrast, the pathway also plays a negative role in some cancers, such as B-cell malignant tumors [27], squamous cell carcinoma [28] and neuroblastoma [29], among others.

## 4. Clinical Significance of Dysfunction of the NOTCH Signaling Pathway in Osteosarcoma Signaling

The clinical significance of the dysfunction of the NOTCH signaling pathway in osteosarcoma was confirmed by the results of many published studies involving several proteins of the pathway signaling (Table 1). A study found that the transcription of NOTCH1.Jag1 and target genes (Hes1 and Hey2) was upregulated in osteosarcoma specimens [30]. The high expression of Hes1 in patients with osteosarcoma is usually associated with a poor survival rate [17]. In addition, a study reported that the low expression of NOTCH1 significantly correlated with low sensitivity to cisplatin in osteosarcoma specimens [31]. Analysis of 70 osteosarcoma specimens revealed that the high expression of NOTCH3 significantly correlated with a low survival rate of patients [32]. Furthermore, multivariate analysis revealed that NOTCH3 was an independent prognostic factor for osteosarcoma [32]. In addition, a study on 68 clinical specimens reported that the NOTCH ligand Jag1 could activate a variety of NOTCH receptors, and its high expression was closely related to the metastasis and recurrence of osteosarcoma [16]. A cohort study of 12 patients with osteosarcoma revealed that the NOTCH1 signaling pathway was significantly upregulated in tumor tissues, and the high expression of the NOTCH1 intercellular domain (NICD1) and the NOTCH target gene Hes1 was associated with a poor response to chemotherapy [33]. However, the expression and clinical significance of NOTCH2 and NOTCH3 in osteosarcoma have not been reported.

In conclusion, the results showed that the NOTCH signaling pathway played an important role in promoting osteosarcoma, and its abnormal activation rather than inactivation accelerates the malignant progression. Therefore, evaluating its expression level and functional status might be significant in predicting the development and prognosis of osteosarcoma.

## 5. Effect of the NOTCH Signaling Pathway on Osteosarcoma

*5.1. The NOTCH Signaling Pathway Regulates Osteogenic Differentiation in Osteosarcoma*

There is a potential link between the occurrence and progression of osteosarcoma and bone differentiation defects [11,12]. Thus, regulating the process of osteogenic differentiation can be considered a promising therapeutic strategy for osteosarcoma [34–36]. Whether NOTCH signaling plays a positive or negative role in osteogenic differentiation remains controversial. In bone marrow mesenchymal stem cells, activated NOTCH signals synergistically enhance the osteogenic differentiation process by increasing the transcription of NOTCH target genes Hes1 or Hey1 and interacting with Runx2 [37], which helps to maintain its phenotype and promote its expansion [38]. On the other hand, NOTCH signaling can also participate in regulating osteogenesis and bone resorption in differentiated bone tissues [38]. Specifically, NOTCH signaling can not only indirectly regulate the differentiation of osteoclasts by regulating the expression of RANKL and OPG in osteoblasts [38] but also directly affect bone resorption by regulating the generation of osteoclasts [37]. A study found that the expression of the NOTCH pathway receptors in various stages of osteogenic differentiation of the human osteosarcoma cell line MG63 was time dependent [39]. The expression of NOTCH1 and NOTCH3 receptors decreased significantly in the early stage, and that of NOTCH2 and NOTCH4 receptors increased significantly in the late stage of osteogenic differentiation [39]. Therefore, the NOTCH pathway plays a dual role in the osteogenic process of MG63 cells, wherein the expression of NOTCH2, NOTCH4, and Hey1

promotes osteoblast differentiation, while the expression of NOTCH1, NOTCH3, and Hes5 maintains the undifferentiated state of osteoprogenitor cells [39]. In genetically engineered mice over-expressing NICD in osteoblasts, 100% of the surviving animals had spontaneous bone tumors [40]. Additionally, most of the mice were diagnosed with high-grade osteosarcoma, which indicated that the abnormal activation of the NOTCH signaling pathway in osteoblasts was significantly related to the occurrence and malignant progression of osteosarcoma [40]. Therefore, the abnormal activation or inhibition of NOTCH signaling leads to the dysregulation of osteogenic differentiation, which may be one of the important mechanisms for the malignant progression of osteosarcoma.

### 5.2. The NOTCH Signaling Pathway Maintains the Self-Renewal Ability of Cancer Stem Cells in Osteosarcoma

Cancer stem cells (CSCs), a cancer cell type with static function and strong self-renewal ability, are located in the primary tumor tissue niche [41]. The existence of CSCs is closely related to chemotherapy resistance, relapse, and metastasis of various cancers [42,43]. NOTCH signaling is associated with the sustained self-renewal ability of CSCs in several malignancies [44,45]. A study found that inhibition of NOTCH1 signaling significantly restrained the growth of CSCs in osteosarcoma [46,47]. Exosomes from human umbilical vein endothelial cells promoted the self-renewal of CSCs in osteosarcoma by enhancing the expression of NOTCH1, Hes1, and Hey1 while blocking NOTCH signaling reversed the positive effect on the CSCs [48]. In addition, studies have reported that the activation of NOTCH1 signaling induced by cisplatin promoted the activity of CSCs in osteosarcoma in vitro, including increasing the number of Stro-1+/CD117+ double-positive cells and spheroid formation capacity [49]. In contrast, inhibiting NOTCH signaling could eliminate these characteristics and decrease the enrichment of osteosarcoma stem cells [49]. In conclusion, the abnormal activation of NOTCH1 signaling exerts a profound effect on the self-renewal of CSCs in osteosarcoma.

### 5.3. The NOTCH Signaling Pathway Promotes Proliferation and Inhibits Apoptosis in Osteosarcoma

Continuous proliferation is one of the prominent characteristics of cancer cells [50]. Dysregulation of the cell cycle is critical for the occurrence of malignant proliferation in cancer [51]. The NOTCH signaling pathway mediated by Jag1 has been proven to be involved in cell cycle regulation in breast cancer [52] and colorectal cancer [53]. A study found that inhibition of the Jag1/NOTCH1 pathway resulted in the arrest of the G1 phase of the cell cycle in osteosarcoma by reducing the expression of cyclin D1, cyclin E1, E2, and Skp2 and promoting the expression of p21 [54]. A study found that the downregulation of Jag1 could significantly inhibit the proliferation of osteosarcoma F5M2 cells; however, the specific mechanism was not explored [16]. Another study showed that the activation of NOTCH signaling mediated by Jag1 promoted the proliferation of K7M2 cells by increasing ERK phosphorylation [55]. Constitutive activation of NOTCH1 signaling can act independently as well as synergistically to promote cell proliferation in osteosarcoma with p53 deletion [40]. In addition to interfering with the cell cycle, NOTCH1 signal activation can promote osteosarcoma cell proliferation by upregulating ephrin1 and enhancing Eph/ephrin reverse signal transduction [33]. Activation of NOTCH1 signaling can also inhibit the apoptosis of osteosarcoma cells by downregulating p21 and Bax and upregulating BCL-2 and BCL-xL [56–58]. Furthermore, NOTCH2 signal activation can promote the proliferation of osteosarcoma cells by upregulating NICD2 and Hes1, while blocking NOTCH2 can inhibit the proliferation of osteosarcoma cells by restraining the progression of the G0/G1 phase of the cell cycle [59].

**Table 1.** Clinical significance and expression level of NOTCH signaling pathway in osteosarcoma.

| Component | Gene | Study/Reference | Functional Status or Expression Level | Clinical Outcome | Cases | Detection Methods |
|---|---|---|---|---|---|---|
| Ligands | Jag1 | [30] | Upregulated | No report | 10 | RT-PCR |
| | | [16] | Upregulated | Increased metastasis rate and recurrence rate | 68 | IHC |
| | | [54] | Upregulated | No report | 10 | RT-PCR |
| | Dll1 | [54] | Downregulated | No report | 10 | RT-PCR |
| Receptors | NOTCH1 | [33] | Upregulated | Reduced cisplatin sensitivity; lower overall survival | 12 | IHC |
| | | [30] | Upregulated | No report | 10 | RT-PCR |
| | | [54] | Downregulated | No report | 10 | RT-PCR |
| | | [31] | High heterogeneity | Positively correlated with cisplatin sensitivity | 8 | IHC |
| | NOTCH2 | [54] | Upregulated | No report | 10 | RT-PCR |
| | NOTCH3 | [32] | Upregulated | Lower survival rates; increased metastasis rates | 70 | IHC |
| Downstream targets | Hes1 | [33] | Upregulated | Reduced cisplatin sensitivity; lower overall survival | 12 | IHC |
| | | [17] | Upregulated | Decreased survival rates | 16 | RT-PCR |
| | | [30] | Upregulated | No report | 10 | RT-PCR |
| | Hey1 | [54] | Upregulated | No report | 10 | RT-PCR |
| | Hey2 | [30] | Upregulated | No report | 10 | RT-PCR |
| | | [54] | Upregulated | No report | 10 | RT-PCR |

### 5.4. The NOTCH Signaling Pathway Promotes Tumor Metastasis and Invasion

Tumor metastasis is a complex and multi-step process controlled by multiple genes that require the cooperation of different molecules [60]. Studies have shown that the NOTCH signaling pathway plays an important role in various steps of the tumor metastasis process in many cancers [61–64]. A study found that the osteosarcoma LM-7 cell line, which showed highly invasive and metastatic features, markedly upregulated the expression of NOTCH1, NOTCH2, Dll1, and Hes1 compared with normal human osteoblasts and the Saos-2 cell line with low metastatic potential [17]. Another study proved that F5M2 cells, an osteosarcoma cell line with high metastatic potential, significantly increased the expression of Jag1 compared with the low metastatic potential osteosarcoma cell line F4, and knockdown of Jag1 resulted in decreased migration and invasion of F5M2 cells [16]. Jag1-mediated NOTCH1 signaling promoted ERK phosphorylation, resulting in increased proliferation and migration of K7M2 cells, while non-selective NOTCH inhibitors significantly inhibited the invasion and migration of osteosarcoma cells by decreasing ERK phosphorylation [55]. Blocking NOTCH1 signaling could effectively reverse the EMT phenotype of osteosarcoma cells induced by low concentrations of DDP, resulting in the attenuation of migration and invasiveness [65]. A study found that NOTCH3 could also promote the invasion and metastasis of osteosarcoma cells by upregulating the downstream target genes Hes1 and MMP7 [32]. In addition, as mentioned above, the NOTCH signaling pathway has positive effects on the recurrence and metastasis of osteosarcoma by maintaining the self-renewal ability of CSCs.

### 5.5. The NOTCH Signaling Pathway Promotes Tumor Angiogenesis

The tumor vascular network, characterized by immaturity, abnormal hyperplasia, and non-functional high-density malformations, constitutively expresses pro-angiogenic factors and provides nutrition for the sustained proliferation of the tumor as well as a channel for distant metastasis [66]. A study reported that the overexpression of VEGF, a key mediator of tumor angiogenesis, is closely related to the low survival rate of patients with osteosarcoma [67,68]. Therefore, targeting VEGF to inhibit tumor angiogenesis is an effective strategy for treating cancer [69,70]. Recent studies have reported that the NOTCH

signaling pathway participates in the regulation of angiogenesis [71]. Inhibition of Dll4-mediated NOTCH signaling in tumors leads to the excessive generation of non-functional vessels [72,73], while the activation of Jag1-mediated NOTCH signaling in vascular endothelial cells promotes tumor angiogenesis [73]. Many studies have shown that VEGF regulates tumor angiogenesis by interacting with the NOTCH signaling pathway. Dll4 is released in the process of vascular sprouting driven by VEGFR2 and NRP1 receptors and, in turn, is part of the negative feedback loop downregulating VEGFR2 and NRP1 on endothelial cells [74,75]. Furthermore, Jag1-mediated NOTCH1 signal activation increased the number of endothelial tip cells, sprouts, and branches at the vascular front by antagonizing Dll4 [76]. Although there are few studies on osteosarcoma [77], NOTCH signaling might regulate tumor angiogenesis by coordinating with VEGF, thus profoundly affecting osteosarcoma.

### 5.6. The NOTCH Signaling Pathway Induces Chemoresistance

The NOTCH signaling pathway plays an important role in tumor drug resistance through multiple mechanisms [78]. As previously mentioned, the low expression of NICD1 and Hes1 in clinical specimens of osteosarcoma was associated with a poor response to chemotherapy [31]. The sensitivity of osteosarcoma Saos-2 cells with a high expression of NOTCH1 to cisplatin was significantly higher than that of MG63 cells with a low expression [31]. In addition, the sensitivity of Saos-2 and MG63 cells to cisplatin was significantly increased by activating NOTCH1 signaling [31]. On the contrary, another study reported that NOTCH1 signaling enhanced chemotherapy resistance by promoting Eph/ephrin reverse signal transduction in osteosarcoma U2OS, MG63, and 143B cells [33]. A sublethal dose of doxorubicin significantly activated the NOTCH signaling pathway, resulting in decreased drug sensitivity in osteosarcoma 143B cells [79]. Similarly, the expressions of NOTCH target genes (Hes1, Hes5, and Hey1, among others) in U2OS and 143B osteosarcoma cells treated with sublethal doses of cisplatin were significantly higher than those of cisplatin-sensitive cells [49]. Low concentrations of DDP induce epithelial–mesenchymal transition (EMT) in osteosarcoma cells by activating the NOTCH signaling pathway, resulting in increased drug resistance, while blocking NOTCH signaling can effectively attenuate the EMT [65].

### 5.7. The NOTCH Signaling Pathway Regulates Immune Infiltration in Osteosarcoma Environment

Immune infiltration, one of the main regulatory factors affecting the progression of tumors, involves all immune cells that make up the TME [80]. Studies have shown that NOTCH signaling can modify the tumor microenvironment in many ways, including regulating the activities of macrophages and MDSCs and directly regulating the cytotoxicity of CD8+ T cells [80]. A study reported that TAMs are widely involved in tumor proliferation, invasion, metastasis, angiogenesis, and CSC characteristics in osteosarcoma [81–83]. A study using NOTCH1 knockout mice reported a significant increase in the infiltration of M2-type TAMs in osteosarcoma tissues accompanied by decreased secretion of the Th1-type cytokines and increased secretion of the Th2-type cytokines [84]. The result of imbalance of the secretion of the two types of cytokines and two types of TAMs resulted in the enhancement of a Th2-type inflammatory response [84]. These results indicated that blocking NOTCH1 signaling could promote the growth and immune escape of osteosarcoma by increasing the polarization of TAMs to the M2 phenotype [84].

## 6. NOTCH Signaling in Animal Models of Osteosarcoma

An ideal animal model of cancer is of extreme significance for the understanding of the mechanism of tumor occurrence as well as for the development of new drugs [85]. To date, the commonly used animal model of osteosarcoma is the tumor transplantation model, which can be divided into the xenograft model (human osteosarcoma animal transplantation) and the allograft model based on the different cell line sources of the host and cells (or tissues) [86]. Currently, the model of concern is the emerging genetically

engineered animal tumor model [87]. The following sections review the research progress on the NOTCH signaling pathway in animal models of osteosarcoma.

### 6.1. Animal Model of Spontaneous Osteosarcoma

The autogenous model is significant in studying the pathogenesis and pathological characteristics of osteosarcoma [88]. Spontaneous osteosarcoma models are commonly created in adult dogs because primary canine osteosarcoma has a high incidence rate and progresses rapidly [89]. A study found that the expressions of NOTCH1, NOTCH2, Hes1, and Hey1 increased markedly in primary canine osteosarcoma [90]. Furthermore, this study also showed that 61 dogs with primary osteosarcoma who underwent amputation and standard chemotherapy revealed that overexpression of Hes1 in tumor tissues was significantly associated with increased disease-free survival [90]. The results indicate that NOTCH signal activation plays an important role in the advancement of canine osteosarcoma through the overexpression of Hes1.

### 6.2. Animal Transplantation Model of Human Osteosarcoma

Considering the existence of species immune rejection between tumor cells and animal transplantation models, immune-deficient nude mice are usually selected, which currently represent the mainstream animal model for osteosarcoma [86]. Blocking the NOTCH signaling pathway inhibits the occurrence and development of primary tibial tumors by downregulating Hes1 in orthotopic xenograft osteosarcoma mouse models [17]. The significant upregulation of the NOTCH1 target gene, Hes1, significantly increased the tumor volume, the positivity rate of tumor Ki67 staining, and lung metastasis in the subcutaneous xenograft mouse model [47]. Moreover, the non-specific activation of the NOTCH signaling pathway promotes CSC characteristics, lung metastasis, and tumor recurrence in subcutaneous xenograft osteosarcoma mouse models [33]. Similarly, inhibition of the NOTCH signaling pathway can effectively inhibit tumor growth and lung metastasis and improve overall survival in xenograft models [91].

### 6.3. Animal Transplantation Model of Allogeneic Osteosarcoma

Currently, murine tumor cell lines are commonly transplanted into allogeneic mice to construct allogeneic tumor transplantation models. K7M2 is a spontaneous osteosarcoma cell line derived from BALB/C mice and is a suitable cell source for allogeneic transplantation. Inhibition of NOTCH signaling was reported to reduce the phosphorylation of ERK, effectively inhibit the growth of osteosarcoma and reduce tumor angiogenesis in tibial osteosarcoma mouse models transplanted with K7M2 cells [55]. Meanwhile, treatment with a NOTCH inhibitor significantly reduced the lung metastasis rate and significantly improved the overall survival rate in the mouse models of osteosarcoma established by tail vein injection of K7M2 cells [55]. In addition, a subcutaneous tumorigenesis model of nude mice was constructed with K7M2 cells. Overexpression of NICD1 significantly increased the tumor volume and the positivity rate of Ki67 staining, thus promoting the proliferation of osteosarcoma cells and enhancing tumorigenicity [33].

### 6.4. Genetically Engineered Animal Osteosarcoma Model

The study of genetically engineered animals is helpful in revealing the molecular mechanisms and development of tumorigenesis. The genetically engineered osteosarcoma animal models often selectively silence the tumor suppressor genes p53 or Rb [92,93]. Bone tumors developed spontaneously in all surviving animals, which were commonly diagnosed as high-grade osteosarcoma in the genetically engineered mice with restricted expression of NICD in osteoblasts [40]. The results indicated that the activated NOTCH signal was significantly related to the occurrence and malignant progression of osteosarcoma [40]. In addition, the aforementioned study also found that abnormal enhancement of NOTCH signaling promoted the tumorigenic effect driven by p53 inactivation, indicating that NOTCH activation can also play a synergistic role with p53 deletion to accelerate

tumor progression [40]. In addition, a study crossed mice with a p53-specific deletion in osteoblasts with genetically engineered mice and suppressed the NOTCH pathway [91]. The study also showed that inhibition of NOTCH signaling attenuates the carcinogenic effect of antagonizing p53 inactivation, reduces the growth and metastasis of osteosarcoma cells, and improves overall survival [91].

## 7. Osteosarcoma Treatment Strategy Based on NOTCH Signaling

The NOTCH pathway, a potential target for tumor therapy, is actively involved in tumor growth, metastasis, chemoresistance, tumor immunity, and other functions. The current therapeutic strategies mostly inhibit the NOTCH pathway to exert antitumor effects [94]. There are mainly two types of NOTCH inhibitors: selective and non-selective. Selective inhibitors include the application of antisense RNA, interfering RNA, and monoclonal antibodies, while non-selective inhibitors include ligand-blocking agents, γ- Secretion inhibitors, and some natural compounds. Selective inhibitors have strong specificity, minimal side effects, and do not easily induce drug resistance. Non-selective inhibitors are more toxic; however, considering the diversity of the NOTCH pathway in cancers, these inhibitors have more clinical value in some cases.

### 7.1. The Biological Agents Targeting the NOTCH Signaling Pathway

#### 7.1.1. Monoclonal Antibodies

Monoclonal antibodies are being developed to block the NOTCH receptors [95]. One type is directed at the extracellular region to block the cleavage of ADAM (a disintegrin and metalloprotease) proteins, and the other type interferes with the binding of the NOTCH ligand and receptor. Monoclonal antibodies blocking NOTCH1, NOTCH2, and NOTCH3 are being explored in clinical trials for the treatment of a variety of tumors [95]. NOTCH1 monoclonal antibody (NRR1) has been used for the treatment of breast cancer, colon cancer [96], and leukemia [96,97], while NOTCH2 monoclonal antibody (NRR2) has been used for the treatment of breast and colon cancer [96], among other cancers. The human monoclonal antibody OMP-59R5 can effectively block NOTCH2 and NOTCH3 signals and is being used pre-clinically for chemotherapy in patients with metastatic or recurrent solid cancers [98,99]. The Dll4 monoclonal antibody (OMP-21M18) has been used to treat colorectal, small-cell lung, pancreatic, and solid cancers [100].

Demcizumab is a human monoclonal antibody against Dll4 [101]. In a clinical study on advanced solid tumors, demcizumab showed a suitable antitumor effect: five patients with sarcoma showed prolonged disease stabilization time and reduced tumor volume after treatment with an appropriate amount [101].

OMP-5948 is a novel cross-reactive antibody that can selectively bind NOTCH2 and NOTCH3 receptors and inhibit signal transduction [98]. In a clinical study of 42 subjects with solid tumors, six patients with sarcomas showed a lower incidence of adverse reactions and better antitumor therapeutic effects after treatment with an appropriate amount of OMP-5948 [98].

Although monoclonal antibodies have rarely been studied in osteosarcoma, they have excellent application prospects in the treatment of other cancers.

#### 7.1.2. Blocking Peptides

SAHM1 is an α-helical polypeptide segment that prevents the assembly of the NICD-MAML-RBPj nuclear complex, abolishing downstream signal transduction by inhibiting target gene transcription [102]. As a NOTCH signal antagonist, SAHM1 effectively inhibits the development of leukemia in mouse models [102]; however, there are no relevant studies on osteosarcoma.

### 7.2. γ-Secretase Inhibitors

γ-secretase, a multi-subunit protein with a complex molecular structure, can hydrolyze a variety of type I transmembrane proteins, acting as a key enzyme in the NOTCH signaling

pathway [103]. In the past, γ-secretase inhibitors (GSIs) were mainly used in the research of Alzheimer's disease [103]. Recent studies have reported that GSIs have an obvious antitumor effect on various cancers [94]. GSIs can specifically inhibit the activity of γ-secretase and reduce the release of NICD, thereby inhibiting the activation of the NOTCH pathway [94,104,105]. Several GSIs have entered the clinical development stage for the treatment of tumors [95].

N-[N-(3,5-difluorophenacetyl)-1-alanyl]-S-Phenylglycinet-butylester (DAPT) is a widely used GSI. Confocal Raman microscopy revealed that DAPT induces nuclear fragmentation and apoptosis in osteosarcoma K7M2 cells and decreases the contents of intracellular nucleic acids, proteins, and lipids, thus inhibiting proliferation and promoting apoptosis in a dose-dependent and time-dependent manner [106]. In addition, it was reported that DART synergized with cisplatin, enhancing apoptosis induced by cisplatin alone in osteosarcoma treatment. It was shown that the downregulation of NOTCH signaling, together with cisplatin, also had a strengthened inhibitory effect on the proliferation and metastasis of the osteosarcoma cell. Moreover, the treatment of osteosarcoma by the combination caused the depletion of the CSCs, thus sensitizing drug-resistant osteosarcoma cells to the cisplatin treatment signaling [107,108]. The results of these studies suggested that the combination of cisplatin and DAPT might be effective and promising for the treatment of advanced osteosarcoma.

RO4929097 is a novel GSI [109]. In a pre-clinical animal trial targeting solid tumors, treatment with RO4929097 resulted in a significant increase in event-free survival in 26 solid tumor xenograft animal models [110]. Compared with other tumor types, RO4929097 had the most consistent and significant growth inhibitory effect in the osteosarcoma group [110]. In a clinical trial on refractory metastatic or locally advanced solid tumors, RO4929097 was found to have suitable drug safety in patients with sarcoma, a satisfactory therapeutic effect, and could effectively prolong the stable disease time [111].

### 7.3. Natural Products

More and more natural products and their extracts have been found to inhibit the NOTCH pathway to exert anticancer effects.

Diallyl trisulfide (DATS), a compound extracted from the allium vegetable [112], has antitumor activity in various cancers [113–115]. A study found that DATS inactivates NOTCH1 signaling to inhibit the proliferation, invasion, and angiogenesis of osteosarcoma cells [77]. These effects were associated with reduced expression of NOTCH-1 and its downstream genes, such as vascular endothelial growth factor and matrix metalloproteinases [77].

Oleanolic acid (OA), a natural triterpenoid compound, has antitumor activity in a variety of tumors [116–118]. A study found that OA could inhibit NOTCH signaling, resulting in promoting mitochondrial apoptosis and inhibiting proliferation in a dose-dependent manner in osteosarcoma [57].

Curcumin is a natural phenolic compound with antitumor activity in different cancers [119–121]. Curcumin has been found to inhibit the proliferation and invasion of osteosarcoma, which is related to the inhibition of the NOTCH-1 signaling pathway [122], but the specific mechanism has not been clarified.

Cinobufacin, a nutmeg steroid extracted from the skin secretion of Bufo gargarizans [123], has antitumor activity against lung cancer [124], breast cancer [125], pancreatic cancer [126], and colon cancer [127]. A recent study has found that cinobufacin can induce apoptosis of osteosarcoma cells by downregulating the expression of NOTCH-1 and its target genes Hes1, Hes5, and Hey1 [56].

The part summarized the clinical application of NOTCH-targeted therapy for the treatment of osteosarcoma (Table 2).

**Table 2.** Osteosarcoma treatment strategy based on NOTCH signaling.

| Drug | Target | Study Types | Study/Reference |
|---|---|---|---|
| Demcizumab | Dll4 | Clinical trial phase I | [101] |
| OMP-5948 | NOTCH2 and NOTCH3 | Clinical trial phase I | [98] |
| DAPT | γ-secretase | Studies performed in in vitro cells and pre-clinical animal models | [106–108] |
| RO4929097 | γ-secretase | Trial performed in pre-clinical animal models | [110] |
| | | Clinical trial phase I | [111] |
| Diallyl trisulfide | A natural product (non-selective inhibitors) | Studies performed in invitro cells | [77] |
| Oleanolic acid | A natural product (non-selective inhibitors) | Studies performed in invitro cells | [57] |
| Curcumin | A natural product (non-selective inhibitors) | Studies performed in invitro cells | [122] |
| Cinobufacin | A natural product (non-selective inhibitors) | Studies performed in invitro cells and pre-clinical animal models | [56] |

## 8. Discussion and Future Directions

NOTCH signaling demonstrates complex and variable effects in distinct malignancies, including tumor-promoting and tumor-suppressing effects. Clinical data confirmed that the NOTCH pathway is activated and highly expressed in osteosarcoma and is closely related to metastasis, drug resistance, and recurrence. Therefore, the NOTCH pathway can be a potential biomarker to predict the prognosis of patients with osteosarcoma. Studies have reported that the effects of the NOTCH pathway varied among different osteosarcoma cell lines, including effects on proliferation, migration, and invasion, the immune system, drug resistance, and cancer stem cell characteristics. In addition, NOTCH signaling plays an important role in the tumor microenvironment, angiogenesis, and osteogenic differentiation, which indirectly affect the biological behavior of osteosarcoma (Figure 2, Table 3). These results support a novel therapeutic strategy for osteosarcoma targeting NOTCH signaling, which has value and broad application prospects. Therapeutically targeting the NOTCH pathway is generally considered complex; however, clinical trials focusing on inhibiting the pathway have demonstrated significant efficacy in the treatment of osteosarcoma. The NOTCH pathway provides a valuable molecular target for the treatment of patients with osteosarcoma, especially in those with advanced disease with chemoresistance and distant metastasis. However, certain challenges persist in the clinical applications of osteosarcoma therapy targeting the NOTCH signaling pathway. The first issue is that the specific mechanism of the NOTCH signaling pathway in osteosarcoma has not been fully elucidated. Second, the intersections or junctions between the NOTCH signaling pathway and other pathways (such as PI3K/Akt/mTOR, Ras/MAPK, and Wnt/β-catenin, among others) should be explored further to provide key targets for precision tumor therapy. Third, the novel NOTCH inhibitors should have both the drug safety advantages of selective inhibitors and the efficacy of non-specific inhibitors. Furthermore, it is possible to combine novel inhibitors with biological tissue engineering for a more efficient targeted drug delivery system.

In conclusion, a comprehensive understanding of the complex functions of NOTCH in osteosarcoma has just begun, and further in-depth research will help in its translation for the clinical treatment of tumors.

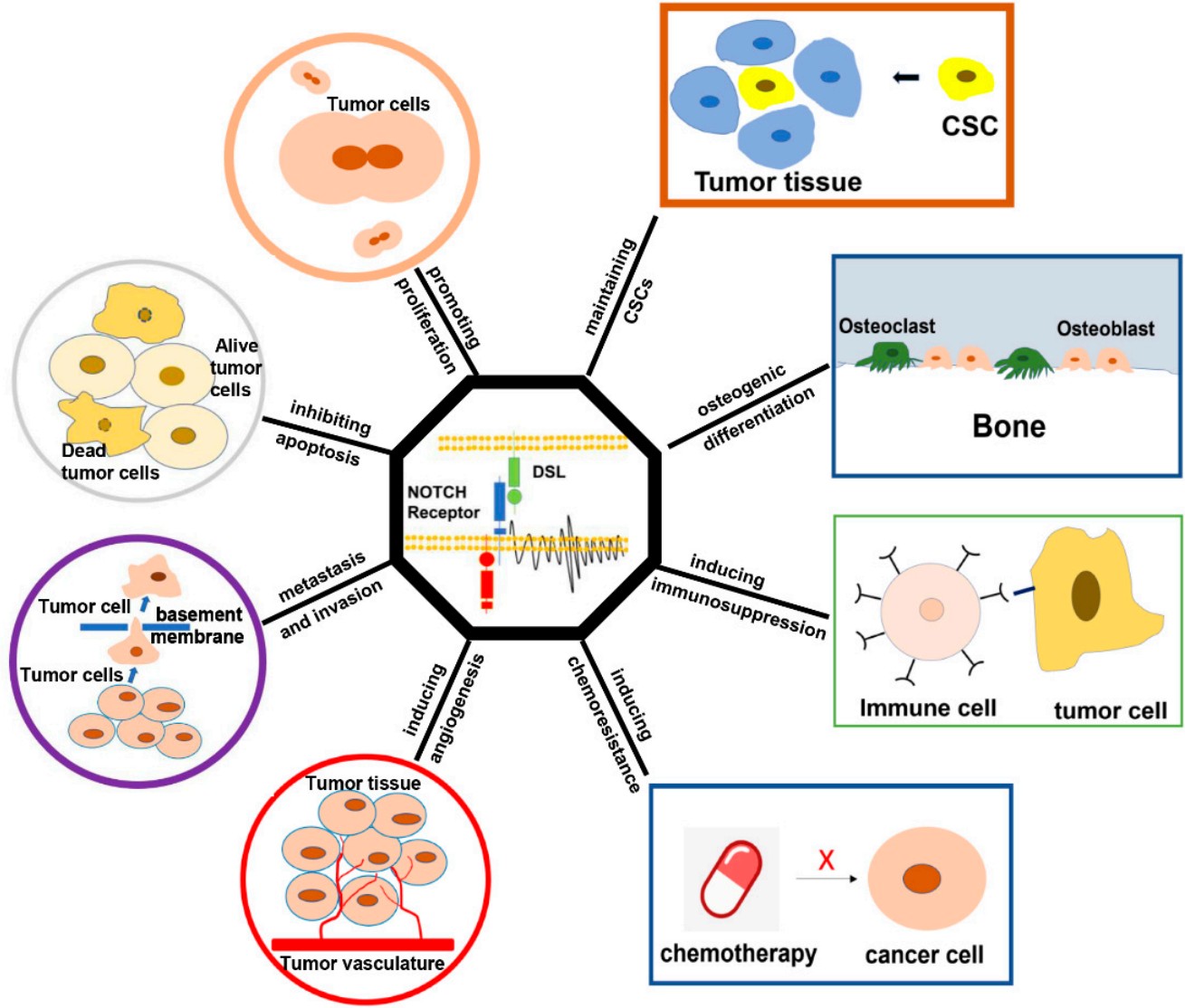

**Figure 2.** Effect of the NOTCH signaling pathway on osteosarcoma.

**Table 3.** Biological effect of the NOTCH signaling pathway on osteosarcoma.

| Component | Gene | Study/Reference | Major Effect | Cell Line | Animal Mode |
|---|---|---|---|---|---|
| Ligands | Jag1 | [16] | Inducing proliferation, migration, and invasion; | F5M2 | No |
| | | [55] | Inducing angiogenesis; promoting tumor growth and metastasis; | K7M2 | Orthotopic allograft mouse model and lung metastatic allograft mouse model |
| | | [91] | Inducing proliferation, migration, and invasion | 143B, SJSA1, SAOS2, U2OS, MG63 | Orthotopic xenograft mouse model |
| | Dll1 | [17] | Inducing proliferation, migration, and invasion; promoting tumor growth and metastasis | OS 187, COL, LM7, SAOS2 | Orthotopic xenograft mouse model |

**Table 3.** *Cont.*

| Component | Gene | Study/Reference | Major Effect | Cell Line | Animal Mode |
|---|---|---|---|---|---|
| Receptors | NOTCH1 | [39] | Inducing osteoblast differentiation | MG63 | No |
| | | [40] | Inducing osteoblast carcinogenesis; inducing proliferation, migration, and invasion; inducing genomic instability | Primary osteosarcoma cells isolated form mice | Genetically engineered mouse model |
| | | [46] | Inducing the activity of cancer stem cell | 143B, MG63 | Subcutaneous xenograft mouse model |
| | | [47] | Promoting tumor recurrence and metastasis | hFOB, SAOS2, MG63, MNNG/HOS, LM5, HuO9, LM132 | Subcutaneous xenograft mouse model |
| | | [48] | Inducing the activity of cancer stem cell | U2OS, 143B | No |
| | | [49] | Inducing the activity of cancer stem cells; promoting tumor proliferation and recurrence | 143B, U2OS, MG63 | Subcutaneous xenograft mouse model |
| | | [55] | Inducing angiogenesis; promoting tumor proliferation and metastasis | K7M2 | Orthotopic allograft mouse model and lung metastatic allograft mouse model |
| | | [33] | Enhancing chemoresistance; inducing the activity of cancer stem cells; inducing proliferation, migration, and invasion; promoting tumor growth, recurrence, and metastasis | U2OS, MG63, 143B | Subcutaneous xenograft mouse model |
| | | [56] | Inhibiting apoptosis | U2OS, MG-63, 143B | Subcutaneous xenograft mouse model |
| | | [57] | Inhibiting apoptosis | SAOS2, MG63 | No |
| | | [58] | Inhibiting apoptosis | MG63 | No |
| | | [77] | Inducing angiogenesis | U2OS, SAOS2, MG63 | No |
| | | [31] | Enhancing chemosensitivity; inhibiting apoptosis | SAOS2, MG63 | No |
| | | [79] | Enhancing chemoresistance | 143B | No |
| | | [84] | Inhibiting the polarization of TAMs to the M2 phenotype | S180, mouse macrophages differentiated by primary bone marrow cells | Subcutaneous xenograft mouse model |
| | | [91] | Inducing proliferation, migration, and invasion | 143B, SJSA1, SAOS2, U2OS, MG63 | Orthotopic xenograft mouse model |
| | | [17] | Inducing proliferation, migration, and invasion; promoting tumor growth and metastasis | OS 187, COL, LM7, SAOS2 | Orthotopic xenograft mouse model |
| | | [30] | Inducing the activity of cancer stem cells; inducing proliferation, migration, and invasion; promoting tumor growth | SJSA1, SaOs2, CRL1423 | Subcutaneous xenograft mouse model |
| | NOTCH2 | [39] | Inducing osteoblast differentiation | MG63 | No |
| | | [59] | Inducing proliferation | 143B, U2OS, MG63, HOS, hFOB | No |
| | | [65] | Attenuating EMT; inducing proliferation, migration, and invasion; promoting tumor growth and metastasis | 143B | Subcutaneous xenograft mouse model |
| | | [17] | Inducing proliferation, migration, and invasion; promoting tumor growth and metastasis | OS 187, COL, LM7, SAOS2 | Orthotopic xenograft mouse model |
| | NOTCH3 | [39] | Inducing osteoblast differentiation | MG63 | No |
| | | [32] | Inducing proliferation, migration, and invasion; promoting tumor growth and metastasis | U2OS, hFOB1.19, MTH | Lung metastatic allograft mouse model |
| | NOTCH4 | [39] | Inducing osteoblast differentiation | MG63 | No |

**Table 3.** *Cont.*

| Component | Gene | Study/Reference | Major Effect | Cell Line | Animal Mode |
|---|---|---|---|---|---|
| Downstream targets | Hey1 | [39] | Inducing osteoblast differentiation; | MG63 | No |
| | | [48] | Increasing the activity of cancer stem cell | U2OS, 143B | No |
| | | [49] | Increasing the activity of cancer stem cells; inducing proliferation; promoting tumor growth and recurrence | 143B, U2OS, MG63 | Subcutaneous xenograft mouse model |
| | | [65] | Attenuating EMT; inducing proliferation, migration, and invasion; promoting tumor growth and metastasis | 143B | Subcutaneous xenograft mouse model |
| | | [79] | Enhancing chemoresistance | 143B | No |
| | Hey2 | [91] | Inducing proliferation, migration, and invasion | 143B, SJSA1, SAOS2, U2OS, MG63 | Orthotopic xenograft mouse model |
| | Hes1 | [46] | Inducing the activity of cancer stem cell | 143B, MG63 | Subcutaneous xenograft mouse model |
| | | [47] | Promoting tumor growth, recurrence, and metastasis | hFOB, SAOS2, MG63, MNNG/HOS, LM5, HuO9, LM132 | Subcutaneous xenograft mouse model |
| | | [48] | Inducing the activity of cancer stem cell | U2OS, 143B | No |
| | | [49] | Inducing the activity of cancer stem cells; promoting tumor growth and recurrence | 143B, U2OS, MG63 | Subcutaneous xenograft mouse model |
| | | [54] | Inducing proliferation; promoting tumor growth | HOS, 143B, SAOS2, U2OS | Subcutaneous xenograft mouse model |
| | | [33] | Enhancing chemoresistance; inducing the activity of cancer stem cells; inducing proliferation, migration, and invasion; promoting tumor growth, recurrence, and metastasis | U2OS, MG63, 143B | Subcutaneous xenograft mouse model |
| | | [17] | Inducing proliferation, migration, and invasion; promoting tumor growth and metastasis | OS 187, COL, LM7, SAOS2 | Orthotopic xenograft mouse model |
| | | [32] | Inducing proliferation, migration, and invasion; promoting tumor growth and metastasis | U2OS, hFOB1.19, MTH | Lung metastatic allograft mouse model |
| | | [79] | Enhancing chemoresistance | 143B | No |
| | | [30] | Inducing the activity of cancer stem cells; inducing proliferation, migration, and invasion; promoting tumor growth | SJSA1, SaOs2, CRL1423 | Subcutaneous xenograft mouse model |

**Author Contributions:** Z.Z. and W.W. wrote the paper and analyzed the data; Z.S. designed the work and revised the manuscript. All authors have read and agreed to the published version of the manuscript.

**Funding:** This research received no external funding.

**Institutional Review Board Statement:** Not applicable.

**Informed Consent Statement:** Not applicable.

**Data Availability Statement:** No new data were created or analyzed in this study. Data sharing is not applicable to this article.

**Acknowledgments:** We would like to express our gratitude to all those who helped us during the writing of this paper. We gratefully acknowledge the help of my supervisor, Zengwu Shao, who has offered me valuable suggestions for academic studies.

**Conflicts of Interest:** The authors declare no conflict of interest.

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
