# Peer review of "NOTCH Signaling in Osteosarcoma"

_cimb, doi:10.3390/cimb45030146_

Round 1

Reviewer 1 Report

Osteosarcoma is the most common primary cancer of the bone. Most osteosarcoma patients currently receive combinatorial treatment of  doxorubicin, cisplatin, and methotrexate as the first-line therapy.

Despite the numerous trials conducted to evaluate novel therapies for metastatic osteosarcoma, the long-term survival of patients remains dismally bad. 

TKIs such as Regorafenib has been the major drug for treating metastatic osteosarcoma .

Other drug classes have been trialed for metastatic osteosarcoma based on promising pre-clinical data but have yielded generally disappointing outcomes.

Therefore, evaluating the potential of therapeutics targeting TORCH for the treatment of the osteosarcoma is of practically clinical importance.

Except the suggested revision commented in the manuscript, there are other two suggestions.

1)   I noticed that the spelling of several  words in the manuscript are in European English. Is it appropriate to alter them to the US spelling , such as signalling to signaling, tumour to tumor, etc.

2)   Is it possible that a table listing all the clinical trials to evaluate the effect of inhibiting NOTCH pathway in osteosarcoma patients?

Reviewer 2 Report

Through this review, the authors describe the roles of Notch signaling pathway in osteosarcomas development and progression, along with the development of potential target therapies.

The review is well written and of interest. My comments are minor, and mostly associated to the format and english. Please review the manuscript again for repetitions and mistakes.

1- The author affiliation is the same for all three authors, except for the email. Would it be possible not to repeat the same affiliation three time and write the emails somewhere else? Two are already written via corresponding author.

2-Line 16, 'different tumors as well as in the same tumor' please rephrase (within the same tumor?)

3- Line 20-21 ' have reported that the NOTCH signalling affected the biological behavior of osteosarcoma the effects through various molecular mechanisms', please rephrase as it is difficult to understand.

4- Line 36 ' the expression level and signalling intensity' What do you mean by signaling intensity? I'm not sure to understand. Isn't it similar (mRNA and protein expression levels are measured via 'intensities' (qPCR, IHC)... isn't it redundant?

5- Line 49, DLL3 is in high caps whereas all other are low caps, is it volunteer? 

6- Line 52, 'encoded by different genes and differ in structure, can be degraded by a variety of proteases'. Would suggest: 'encoded by different genes, differ in structures and can be degraded by a variety of proteases'

7- Line 70, same comment concerning signal intensity.

8-Line 73 and 74, 'occurence and development' is repeated twice. Maybe reformulate differently?

9- Line 155 and 156, 'one of the' is repeated twice, would it be possible to find an alternative to prevent repetition?

10- Line 158 and 159, 'colorectal cancer' is repeated twice.

11- Line 238, 'reported that' repeated twice.

12-Line 248-249 ' development of treatment methods animal model' I think the 'animal model' at the end is not required. 

13- Line 252-253, 'the emerging' repeated twice.

14- Line 262-264, 'primary osteosarcoma underwent amputation'. I think it would be easier to read if 'who' or 'that' was inserted between osteosarcoma and underwent.2

15- Line 299 ' T The results', please erase the T.

16- Line 301, 'Found that also found that' please correct. 

17- Line 355, there's two spaces between an and obvious

18- Line 365, the comma between signalling and resulted is not required.

19- Line 371, two spaces between survival and in

20- Please review the reference numbers to be consistent throughout the manuscript. Some have spaces between the word and the number, other not. 

Thank you!

Reviewer 3 Report

Kindly find the comments below for revision,

 Major comments

This is not the first review of Notch signalling in osteosarcoma. In the introduction chapter, author should cite below article or similar articles. 

MCMANUS, Madonna M.; WEISS, Kurt R.; HUGHES, Dennis PM. Understanding the role of Notch in osteosarcoma. Current advances in osteosarcoma, 2014, 67-92.

Figure 1 needs more precision. Currently the figure is vague and unclear.

Figure 2 needs more precision. What are the cell types present in each circle? Currently the figure is not self-descriptive.

New table is needed to show all the notch signalling related studies in osteosarcoma. Mention the tissue source, cohort size, protein level or mRNA level when overexpressed, studies performed in invitro cells or pre-clinical animal models, are there any relevant clinical trials? etc.  

Minor comments

check the typo in the keywords “treatmentï¼›

Signalling or signaling,  use the same word throughout.

Dll1, Dll4 and DLL3- uniformity in the abbreviations missing (some letter are in caps and some are not)

Jag 1 or Jagged 1 – if they are same, use the same word throughout.

Sometimes Hey1- sometimes Hey-1. Use the same word throughout.

In line 158 consider rephrasing “involved in cell cycle regulation in breast and colorectal cancers and colorectal cancer”

In line 299 check the typo “T The results I”

In line 351 check the typo “7.2. γ-. secretase inhibitors”

In line 417 consider rephrasing “the NOTCH signalling pathway, thereby necessitating scientific research and clinical personnel”

AD-AM protease- full name of the abbreviate AD-AM is need. Check for all abbreviation in the manuscript.  

Round 2

Reviewer 3 Report

Authors have improved the manuscript by adding tables and citing the relevant articles.

However, still figure 1 needs to be improved extensively.

Also in table 3, mention the exact/precise cell lines and animal models used in those cited studies. Currently, it is vague and unclear. 

Author Response

Point 1: still figure 1 needs to be improved extensively.

Response 1: Thanks. Figure 1 has modified to be clearer as your suggestions.

Point 2: Also in table 3, mention the exact/precise cell lines and animal models used in those cited studies. Currently, it is vague and unclear. 

Response 2: Thanks. Table 3 has modified as your suggestios.